# Optical Behavior and Surface Analysis of Dental Resin Matrix Ceramics Related to Thermocycling and Finishing

**Liliana Porojan** [1,*] , **Flavia Roxana Toma** [1], **Ion-Dragoș Uțu** [2] **and Roxana Diana Vasiliu** [1,*]

1   Department of Dental Prostheses Technology (Dental Technology), Center for Advanced Technologies in Dental Prosthodontics, Faculty of Dental Medicine, "Victor Babeș" University of Medicine and Pharmacy Timișoara, Eftimie Murgu Sq. No. 2, 300041 Timișoara, Romania; flavia.toma@umft.ro

2   Department of Materials and Fabrication Engineering, Politehnica University Timișoara, Bd. Mihai Viteazul nr.1, 300222 Timişoara, Romania; dragos.utu@upt.ro

*   Correspondence: sliliana@umft.ro (L.P.); roxana.vasiliu@umft.ro (R.D.V.)

**Abstract:** Color preservation of esthetic dental restorative materials in the oral environment represents, besides longevity, a concern, and there is still limited knowledge related to the effect of aging on the optical behavior of resin matrix ceramics. The study analyzed the finishing and thermocycling of resin matrix ceramic material surfaces, in order to assess their consequences on optical properties. Five resin matrix CAD/CAM ceramics, namely a polymer-infiltrated ceramic and four types of nanoparticle-filled resins, were selected for the study, and finished by polishing and glazing. Thermocycling was chosen as the in vitro aging method. Surface microroughness, optical and hardness evaluations were achieved before and after artificial aging. Statistical analyses were performed with IBM SPSS Statistics software at a significance value of $p < 0.05$. Micro-roughness values increased after thermocycling, but were kept under the clinically accepted values. The optical characteristics of resin matrix ceramics were not significantly modified by thermocycling. Values of the glazed samples became closer to those of the polished ones, after hydrothermal aging, even if the differences were insignificant. Thermocycling significantly decreased the microhardness, mainly for glazed samples. This could be the consequence of glaze removal during thermocycling, which means that glazes provide a surface protection for a limited time.

**Keywords:** resin matrix ceramics; thermocycling; optical properties; surface analysis

## 1. Introduction

Related to the high demand of patients for biomimetic dental restorative materials, ceramics and composites come into question. Longevity is an essential aspect when restoring natural teeth structures and it is important that the physical properties of the restorative materials overlap the tooth structures [1–3]. The rapid development of CAD/CAM technologies provided accurate, repeatable and high-quality dental restorations using a variety of materials. CAD/CAM systems represent an outstanding advancement, offering practitioners the chance to achieve improved dental restorations [4,5].

During the last decade, a new category of dental esthetic materials has been researched, called resin matrix ceramics or hybrid ceramics. These materials consist of an organic matrix with a high content of ceramic particles. These materials are included in the classification of dental ceramics, after the American Dental Association Code on Dental Procedures and Nomenclature (version 2013), which defines the term porcelain/ceramic as "pressed, fired, polished, or milled materials containing predominantly inorganic refractory compounds—including porcelains, glasses, ceramics, and glass-ceramics". The former version of the ADA code did not include materials with resin matrix in the category of ceramic materials [6–9].

The physical properties related to resin matrix ceramics lie very close to those of natural teeth, and between ceramics and composites. The milling time in the CAM unit

is shorter compared to other ceramic materials, leading to a longer lifetime of the milling burs. The polymer structure of the material could also prevent crack propagation through the ceramic phase. Well-known resin composites wear out; they lose their surface gloss and color stability overtime more easily compared to ceramic materials [2,10–20].

As surface processing methods of these materials, conventional polishing and glazing for high gloss and gloss stability are designated. Glaze materials are used to seal the surfaces and to obtain smoother and glossy surfaces. This has a great impact in decreasing surface asymmetries, increasing wear resistance, and improving the resistance to staining [21,22]. The surface roughness of a material is important for esthetics and long-term success, but there is a need for more studies related to the effectiveness of glazes on CAD/CAM resin matrix ceramic materials.

Ensuring success in esthetic restorations is a difficult process, because of the optical complexity of tooth structures. To obtain success, basic principles in optical characterization, such as mimicking of the opalescence and translucence of natural teeth, must be taken into consideration [23–25]. Due to the hydrophilic characteristics of resin-based materials, studies demonstrated the relation with their optical properties. Resin-based materials are known for their esthetic appearance, strength, and accessibility, but lack the strength and optical stability of ceramic materials. Thus, resin matrix ceramics were introduced in order to combine the benefits of ceramics and composites. They may limit the absorption of water compared to resin-based materials, and thus preserve optical properties.

Information regarding the properties of resin matrix ceramic materials after a prolonged usage period is limited [18]. The Vickers hardness test is important because it is a predictor of surface wear [26].

Thermocycling is one of the widely used aging methods to investigate restorative materials and there are different in vitro aging protocols. Thermocycling represents the oral environments better than isothermal storage conditions. Temperatures of 5–55 °C are considered the limits for the real situation in the oral cavity. A total of 10,000 cycles were selected for the study, which represent one year of clinical service and are generally used for in vitro studies [27,28].

Dental restorations are exposed to dynamic temperature changes in the oral cavity. The different thermal expansion coefficients of the filler particles and resin matrix induce internal stresses in the material during continual temperature changes. Resulting cracks and gaps damage the mechanical properties over time, accelerating aging. Additionally, in materials with nano-zirconia filler, zirconia may undergo a low-temperature degradation with a phase transition from tetragonal to monoclinic phase at oral temperature, overlapping the abovementioned degradation [29–31]. It was demonstrated that a definite water content of enamel and dentine structures is mandatory for their physical characteristics and biomechanical behavior, providing stability and robustness. Water absorption induces a greater risk of decomposition of the molecular structure on the microscale, and is considered to change the physical properties and the lifetime of dental restorative materials. Accordingly, the impact of water content on dental materials represents a challenge, as well as their susceptibility to degradation in water and aging [32–34]. Polymer networks can be strongly altered by the wet oral environment [35]. There is still limited information regarding the effect of aging on the optical stability of these materials [36–42] Color preservation of esthetic dental restorative materials in the oral environment represents besides longevity a concern in the use of esthetic dental restorative materials. Restorations should reproduce the optical properties of natural teeth and maintain this natural appearance over time. These are directly related to optical harmony with natural teeth and to the color stability of the material. Spectrophotometers are currently used to evaluate the color change ($\Delta E^*$) of dental materials by quantifying the color coordinates in the CIELAB color space: $L^*$, $a^*$ and $b^*$. The $L^*$ coordinate is an indicator of brightness, the $a^*$ coordinate indicates the red–green component (negative $a^*$ represents greenness and positive $a^*$ indicates redness), and the $b^*$ coordinate corresponds to yellowness or blueness (negative $b^*$ indicates blueness and positive $b^*$ represents yellowness). $L^*$ coordinates are the most important parameters

in the optical appearance of restorations, because the lightness has the greatest effect on esthetics [43–45].

The study aimed to conduct a surface analysis of resin matrix ceramic materials related to finishing and thermocycling, and to assess their consequences on optical properties. The study included hybrid materials that were tested after hydrothermal aging for changes in the optical properties such as translucency and opalescence, and as well for surface and mechanical properties.

The null hypothesis is that the optical and surface properties of the hybrid ceramic included in this study are influenced by the hydrothermal aging.

## 2. Materials and Methods

### 2.1. Specimen Preparation

Five resin matrix CAD/CAM ceramic materials were selected for the study: a polymer infiltrated network ceramic (Vita Enamic VITA Zahnfabrik, Bad Säckingen, Germany) (E) and four types of ceramic nanoparticle-filled resins (Lava Ultimate, 3M ESPE, St. Paul, MN, USA (L), Cerasmart, GC Corporation, Tokyo, Japan (C), Shofu HC, Shofu, Kyoto, Japan (S), and Hyramic Upcera, Liaoning, China (H)) were selected for the study (Table 1), with shades A2 or 2M2 and translucency HT. The CAD/CAM blocks were sliced into rectangular-shaped plates ($n = 16$) per material using a machine (Orthoflex PI Dental, Budapest, Hungary) that provides millimeter accuracy. The samples were polished using silicon carbide papers (600–2000 grit) and the final thickness (1 mm) of each plate was checked with a digital caliper.

**Table 1.** Composition and manufacturer specifications of tested materials [20,45,46].

| Material | Type | Manufacturer | Filler | Monomer | Shade/Translucency |
|---|---|---|---|---|---|
| Vita Enamic (E) | Hybrid ceramic | VITA Zahnfabrik, Bad Säckingen, Germany | Feldspar ceramic enriched with aluminum oxide 86% | UDMA, TEGDMA | A2/MT |
| Lava Ultimate (L) | CAD/CAM composite resin | 3M ESPE, Seefeld, Germany | $SiO_2$, $ZrO_2$, aggregated $ZrO_2/SiO_2$ cluster 80% | UDMA, Bis-GMA, Bis-EMA, TEGDMA | A2/MT |
| Cerasmart (C) | CAD/CAM composite resin | GC Corporation, Tokyo, Japan | Silica, barium glass 71% | UDMA, Bis-MEPP, DMA | A2/MT |
| Shofu HC (S) | CAD/CAM composite resin | Shofu, Kyoto, Japan | Silica, silicate, zirconium silicate 61% | UDMA, TEGDMA | A2/MT |
| Hyramic (H) | CAD/CAM composite resin | Upcera, Liaoning, China | Inorganic Filler 55–85% | Resin Polymers | |

The surfaces were finally polished with a low-speed handpiece using a diamond polishing paste, Renfert polish all-in-one (Renfert, Hilzingen, Germany). Half of the specimens from each material were kept polished (*p*) and half were glazed (*g*). Resin Glaze Primer (Shofu, Kyoto, Japan) was applied to the surfaces for 60 s and allowed to dry. After that, two thin layers of the glaze Resin Glaze Liquid (Shofu, Kyoto, Japan) were applied with a soft brush in the same direction in order to eliminate air bubbles and were light cured for 180 s in the device SibariSr 620 (Sirio Dental, Meldola, Italy). After the surface finishing protocols, all plates were ultrasonically cleaned for 10 min and degreased in alcohol.

### 2.2. Hydrothermal Aging Protocol

Prior to baseline measurements, all samples were immersed in distilled water at 37 °C for 24 h to allow full hydration. Further specimens were submitted to 5000 cycles in baths filled with distilled water at 55 °C and 5 °C. Each cycle lasted 80 s; 30 s was the dwelling time in the 5 °C bath, 10 s were needed for transfer to the other bath, 30 s was the time in the 55 °C bath, and 10 s were needed to transfer of the samples back to the 5 °C bath (t1). The other 5000 cycles were followed using the same protocol (t2).

The hydrothermal aging protocol has an indication for this type of dental materials. This method is used to simulate the in vivo aging of the dental restorative materials by

subjecting them to repeated cold and warm water baths [47]. These water baths simulate the changes that occur in the oral cavity of the patients.

### 2.3. Surface Microroughness Evaluation

Surface microroughness was measured with a 2 μm contact stylus profilometerSurftest SJ-201 (Mitutoyo, Kawasaki, Japan) before (w) and after each step of aging (t1 and t2). Arithmetic average roughness (Ra) [48,49] evaluations were performed in five different directions, all data were recorded and mean values of the five measurements were calculated for each surface. The used sampling length was 0.3 mm, and a force of 0.7 mN was applied.

### 2.4. Optical and Color Changes Measurements

Translucency (TP) and opalescence (OP) values were calculated for all samples, on both types of surfaces (polished and glazed), before (w) and after each step of aging (t1 and t2). Optical parameters were registered under a D65 illuminant, using a Vita Easyshade IV spectrophotometer (Vita Zahnfabrick, Bad Säckingen, Germany). It was calibrated before each measurement.

To assess the measurements, two different backgrounds, black (b) and white (w), were selected, using the grey card WhiBal G7 (White Balance Pocket Card). L* represents the lightness–darkness of the material (L* = 0 for a perfect black and L* = 100 for a perfect white). a* represents the measure of greenness (negative value) or redness (positive value), and b* measures the blueness (negative value) or the yellowness (positive value) [50–53].

TP values were calculated using Equation (1).

$$TP = [(L_b - L_w)^2 + (a_b - a_w)^2 + (b_b - b_w)^2]^{1/2} \tag{1}$$

OP values were calculated using Equation (2).

$$OP = [(a_b - a_w)^2 + (b_b - b_w)^2]^{1/2} \tag{2}$$

The CR (contrast ratio) value was achieved by Equation (3).

$$CR = Y_b/Y_w \; Y = [(L^* + 16)/116]^3 \times 100 \tag{3}$$

w and b correspond to the white and black backgrounds, respectively. CR = 0 is equivalent to transparent, and CR = 1 to totally opaque [54].

The color changes (ΔE*), which represent the color difference between two stages, were calculated based on the CIE L*a*b*color system, for a black background, according to Equation (4).

$$\Delta E^* = [(\Delta L^*)^2 + (\Delta a^*)^2 + (\Delta b^*)^2]^{1/2} \tag{4}$$

The National Bureau of Standards (NBS) system quantifies the levels of color change, relating to a clinical standard. ΔE* values were thus converted into NBS units: NBS unit = ΔE* × 0.92 [52,55–59].

### 2.5. Microhardness Evaluation

Samples were evaluated for the Vickers microhardness before (w) and after aging (t2). Measurements were made on selected points using the digital camera of the tester, with the micro-hardness tester DM 8/DM 2 (Yang Yi Technology Co., Ltd., Tainan City 70960, Taiwan) using a diamond pyramidal indenter with 300 g load, for 10 s. After lifting the indenter, the indentation dimensions were microscopically recorded (40× magnifications). Five measurements were done on each surface, and mean values were calculated. The formula used for the Vickers microhardness calculation is the following (5):

$$HV = 1.8544 \; F/d^2 \tag{5}$$

where HV is the Vickers hardness value, F is the applied load value and d is the diagonal length of the indentation [60].

### 2.6. Statistical Analysis

IBM SPSS Statistics software (IBM, New York, NY, USA) was selected to achieve statistical analysis. For all studied materials as well as for all surface processing procedures, mean values for color parameters, roughness, surface hardness, SDs and 95% confidence intervals were subsequently calculated related to hydrothermal aging stages. Statistical evaluation for material type and surface processing was performed. Comparison of different aging stages was conducted using a paired samples *t* test. Differences were considered significant if the corresponding *p* value was <0.05. A statistical correlation (Spearman) was used to establish relationships between optical parameters. It measures the strength of correlations between variables and the direction of the relationship. The significance was calculated as follows: 0–0.19 "very weak", 0.20–0.39 "weak", 0.40–0.59 "moderate", 0.60–0.79 "strong", and 0.80–1.0 "very strong".

### 3. Results

Before thermocycling (w), average roughness values (Ra) showed significant differences between polished and glazed samples for L (*p* = 0.035) and H (*p* = 0.033). Glazing increased roughness values in L, C, and S samples and decrease them in E and H. The roughness values increased as follows: Sp < Ep < Lp < Hp < Cp, Hg < Eg < Sg < Cg < Lg. Differences were significant between Cp and Sp (*p* = 0.003), Ep (*p* = 0.014), and Lp (*p* = 0.029), respective Lg and Hg (*p* = 0.036).

Related to different surface finishing methods, after hydrothermal aging polished surfaces had significantly lower roughness than glazed ones for C (*p* = 0.007) and H (*p* = 0.001). The values increased as follows: Hp < Sp < Cp < Ep < Lp, Lg < Sg < Eg < Cg < Hg. Related to thermocycling, Ra values increased significant for Ep, Lp, and Sp, respective Eg, Sg, and Hg (Figure 1).

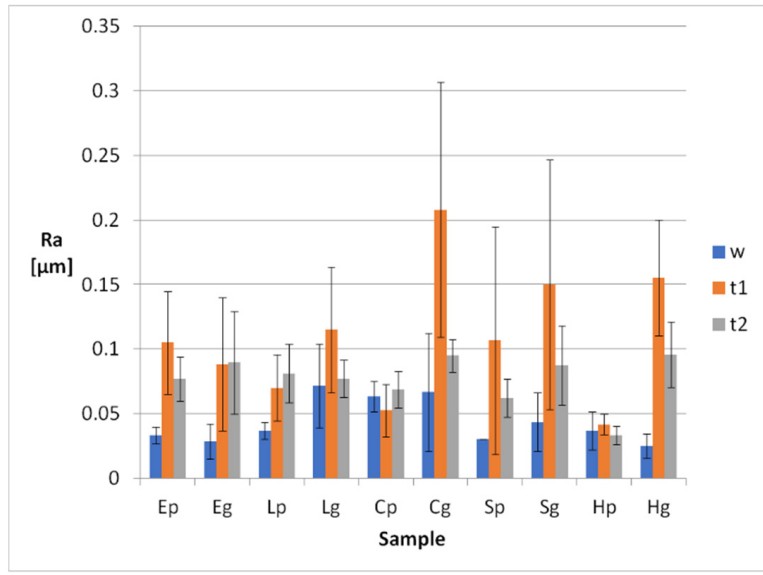

**Figure 1.** Mean Ra roughness values and SD values of the samples, before and after thermal aging.

Related to surface processing, TP and OP values increased and CR decreased for glazed samples, with results being significant before thermocycling and insignificant after.

Mean calculated optical parameters ranged between 17.01 and 24.79 for TP, 4.40 and 8.05 for OP, and 0.46 and 0.60 for CR (Figures 2–4, Tables 2–4). TP, OP and CR values were insignificantly changed by thermocycling.

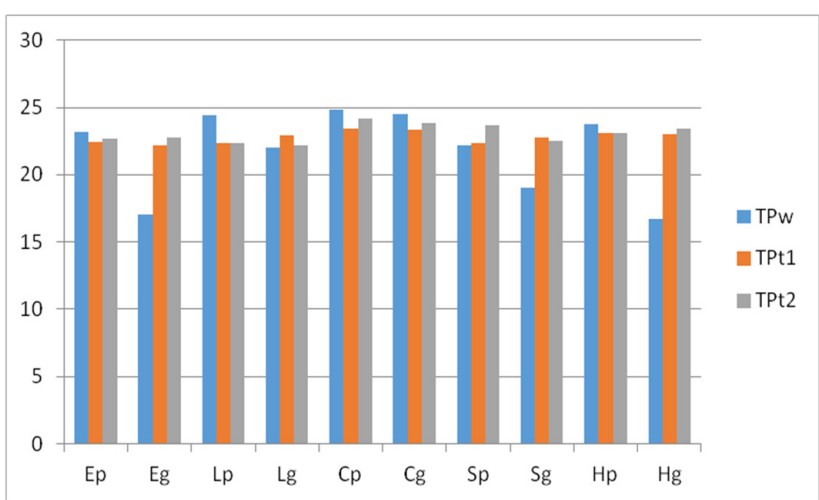

**Figure 2.** Mean values of the TP parameter, before and after thermal aging.

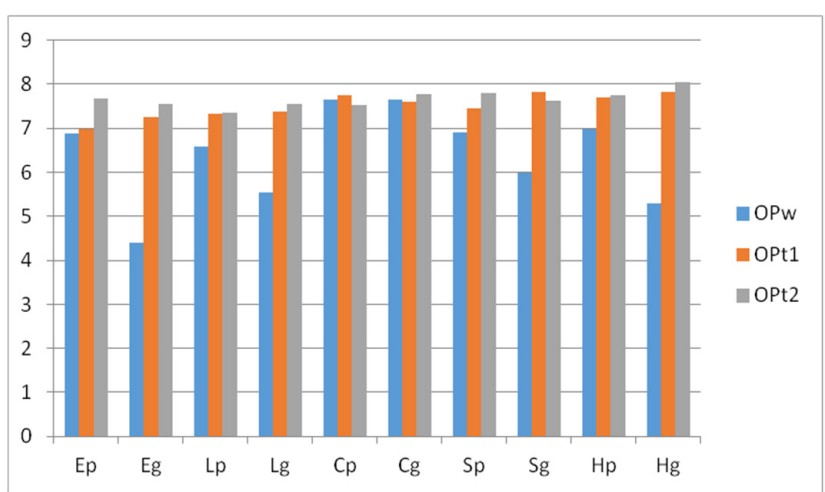

**Figure 3.** Mean values of the OP parameter, before and after thermal aging.

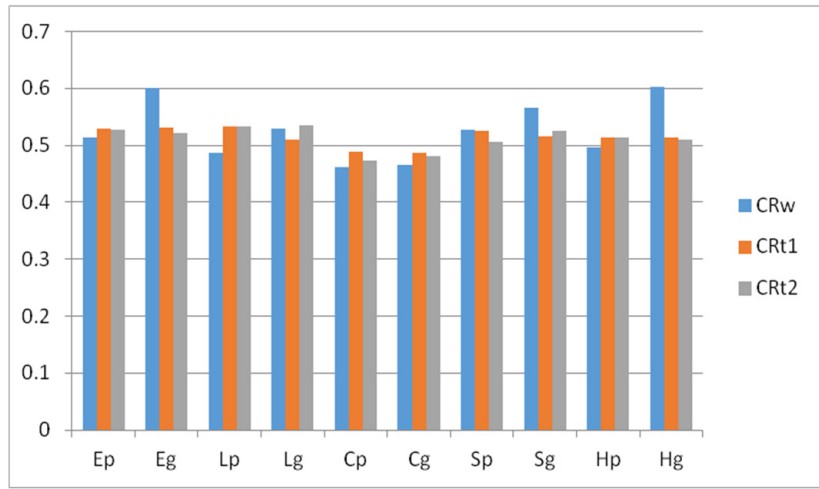

**Figure 4.** Mean values of the CR parameter, before and after thermal aging.

TP and OP values were very strongly positively correlated before thermocycling (0.891), strongly correlated after the first step (0.648), and moderately correlated after the second (0.455). TP and CR values were very strongly negatively correlated before

thermocycling ($-1$), after the first step ($-0.927$), and after the second ($-0.988$). OP and CR values are very strongly negatively correlated before thermocycling ($-0.891$), and moderately correlated after the first ($-0.552$), and after the second thermocycling step ($-0.442$).

**Table 2.** Mean TP values of samples after water immersion, before thermocycling (w), after thermocycling for 5000 cycles (t1) and after thermocycling for 10,000 cycles (t2). *,a,b significant changes after hydrothermal aging.

| Sample | Ep | Eg | Lp | Lg | Cp | Cg | Sp | Sg | Hp | Hg |
|---|---|---|---|---|---|---|---|---|---|---|
| TPw | 23.14 * | 17.01 * | 24.45 [a] | 21.97 [a] | 24.79 | 24.52 | 22.21 | 19.00 | 23.71 [b] | 16.74 [b] |
| TPt1 | 22.42 | 22.19 | 22.32 | 22.95 | 23.41 | 23.33 | 22.38 | 22.74 | 23.13 | 23.03 |
| TPt2 | 22.66 | 22.75 | 22.35 | 22.21 | 24.15 | 23.82 | 23.64 | 22.50 | 23.11 | 23.41 |

**Table 3.** Mean OP values of samples after water immersion, before thermocycling (w), after thermocycling for 5000 cycles (t1) and after thermocycling for 10,000 cycles (t2). *,a significant changes after hydrothermal aging.

| Sample | Ep | Eg | Lp | Lg | Cp | Cg | Sp | Sg | Hp | Hg |
|---|---|---|---|---|---|---|---|---|---|---|
| OPw | 6.89 * | 4.40 * | 6.58 [a] | 5.54 [a] | 7.66 | 7.64 | 6.90 | 5.99 | 6.98 | 5.28 |
| OPt1 | 6.98 | 7.24 | 7.33 | 7.37 | 7.74 | 7.60 | 7.44 | 7.82 | 7.71 | 7.83 |
| OPt2 | 7.67 | 7.56 | 7.34 | 7.55 | 7.53 | 7.78 | 7.80 | 7.62 | 7.76 | 8.05 |

**Table 4.** Mean CR values of samples after water immersion, before thermocycling (w), after thermocycling for 5000 cycles (t1) and after thermocycling for 10,000 cycles (t2).

| Sample | Ep | Eg | Lp | Lg | Cp | Cg | Sp | Sg | Hp | Hg |
|---|---|---|---|---|---|---|---|---|---|---|
| CRw | 0.51 | 0.60 | 0.48 | 0.52 | 0.46 | 0.46 | 0.52 | 0.56 | 0.49 | 0.60 |
| CRt1 | 0.52 | 0.53 | 0.53 | 0.51 | 0.48 | 0.48 | 0.52 | 0.51 | 0.51 | 0.51 |
| CRt2 | 0.52 | 0.52 | 0.53 | 0.53 | 0.47 | 0.48 | 0.50 | 0.52 | 0.51 | 0.51 |

The calculated NBS levels of color change registered marked changes (NBS units > 3) only for Lp samples, after thermocycling (t1 and t2).

Microhardness values were significantly lower for *p* than *g* samples before thermocycling, for all resin nanoceramic materials ($p < 0.05$). For E the values were similar ($p = 0.094$). After thermocycling the differences between *p* and *g* remained significant for L and H. Related to thermocycling, the microhardness values decreased significantly for all samples, excepting Sp (Figure 5).

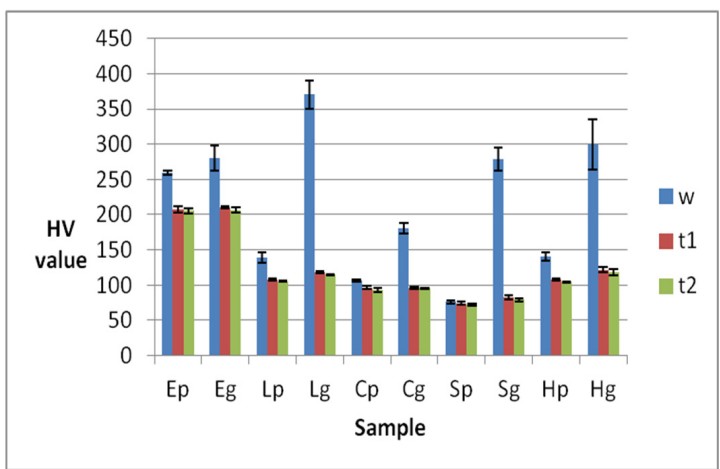

**Figure 5.** Mean microhardness values and SD of the samples, before and after thermocycling.

Related to materials, for *p* samples before thermocycling, the values decreased significantly as follows: Ep > Hp > Lp > Cp > Sp, except from Hp to Lp. For g samples they decreased as follows: Lg > Hg > Eg > Sg > Cg, being significant only between Sg and Cg.

After thermocycling, values decreased as follows: Ep > Lp > Hp > Cp > Sp, being insignificant between Lp and Hp, and Cp and Sp. For glazed samples they decreased as follows: Eg > Hg> Lg > Cg > Sg, being insignificant between Hg and Lg.

## 4. Discussion

Based on the results of this in vitro study the null hypothesis was confirmed, namely that all of the materials included in this study were influenced by the hydrothermal aging.

Based on the results, thermocycling significantly decreased the Vickers hardness. Thermocycling may cause water absorption in the resin material structure. A consequence is an expansion of the network and decreasing of the frictional forces between polymer chains. Studies made suppositions that the assimilated water would lead to hydrolysis of the silane coupling agent, altering the chemical bond between the fillers and the resin matrix [19,61,62]. Other studies speculated that differences in the thermal expansion coefficients of the two phases could induce stress at the network interfaces. Experiments demonstrated that an equilibrium should be reached after 30 days and further storage in water did not alter more the mechanical properties [63].

The hardness of polished samples decreased with the decrease of the inorganic filler content, both before and after thermocycling, and for glazed samples after thermocycling, because the glaze was not kept on the surface. Glaze temporary increased the microhardness, but was removed by thermocycling. This means that aging due to thermocycling affecting the materials differently in relation to surface processing and glazing should be avoided. Other studies showed that different materials behave differently under thermocycling [63]. Related to the studied materials, PICN ceramic (E) was obtained through infiltration of presintered ceramic with resin, inducing a higher density. This could be another reason that supports the higher microhardness of this material. On the other hand, resin nanoceramic materials contain hydrophobic elements such asurethane dimethacrylate (UDMA), or hydrophilic elements such as triethylene glycol dimethacrylate (TEGDMA), and bisphenol A-glycidyl methacrylate (Bis-GMA), which can explain the decreased microhardness caused by an increase of the susceptibility to water sorption [13,63,64]. Ethoxylatedbisphenol A diglycidyl methacrylate (Bis-EMA) is a type of ethoxylatedBis-GMA that is highly hydrophilic and has no reactive hydroxyl group in its main polymer chains. It should therefore exhibit insignificant water sorption. Studies investigated composites with different filler particles and showed that those with larger fillers are more susceptible to color changes related to water sorption than those with smaller filler particles, which is due to the hydrolysis at the filler–matrix interface [19,27,65,66]. All of these previous studies support the results of this study, because hydrophobic elements are included in the matrix of resin nanoceramics and the filler particles size is also small.

Related to microroughness, all Ra values increased after thermocycling, but they were kept under the clinically accepted value, just below 0.1 μm after the second step of thermocycling. Ra values below 0.2 μm are generally clinical accepted [3,67].

Translucency allows passing of light, and it is known as the state between opacity and transparency. Opalescence is related to the light scattering of short wave-lengths from the visible spectrum in translucent materials. This characteristic gives a material a bluish-white aspect in reflected light and an orange–brown aspect in transmitted light. Restorative materials should have similar opalescent properties to natural tooth structures. Among all of the optical properties, translucency is known as a key factor for the natural outcome of esthetic restorations [68–73].

Previous studies registered mean TP values for 1 mm thick human enamel as 18.7 and for human dentine as 16.4 [74–76] The TP values for the studied resin matrix ceramic samples lie between 17.01 and 24.79. The OP value of the enamel–dentin complex was reported to be 4.8, and that of enamel, 7.4 [77–79]. The values for the studied resin matrix

ceramic samples lay between 4.40 and 8.05. The CR and TP values of esthetic dental materials were compared in different studies. CR was negatively correlated to TP (r = −0.93). The calculated correlation was also very strong during all of the tested periods [70,79]. All of the color parameters were insignificantly changed by thermocycling. The values of the glazed samples became closer to those of the polished ones, even if the differences were insignificant between *p* and g samples. This could be a consequence of the glaze being removed after thermocycling. A marked color change was registered only for Lp.

One of the limitations of the investigations is related to the applied aging method, which is limited related to the complex clinical conditions. Another limitation of this study could be the selection of one shade (A2 or 2M2) and translucency (HT) for the tested materials. Considering that besides water absorption and the thermal changes many other factors age dental restorative materials, further studies are needed that evaluate the restorative materials using various and complex aging procedures, in order to simulate as close as possible the real oral environment [18,80,81].

However, in vitro tests can exhibit useful indications for newly developed materials as long as the testing protocol is accurate, the investigations are well-conducted and the results are carefully interpreted, taking into account the limitations [82–84].

## 5. Conclusions

Within the mentioned limitations of this in vitro study, the following conclusions can be drawn:

1. The optical properties of resin matrix ceramic materials are not significantly modified by thermocycling.
2. Roughness values, even if they increase after aging, are kept under the clinically accepted values.
3. Thermocycling significantly decreases the Vickers hardness.
4. Glaze provides surface protection for a short time, because it is removed during thermocycling.
5. This research stands as a starting point for future research because of the various hybrid materials included in this text and of the significant results. Taking everything into consideration, we can only encourage further investigations so that the results of different studies can be compared and analyzed properly.

**Author Contributions:** Conceptualization, L.P. methodology, L.P.; software, L.P.; investigation, I.-D.U., R.D.V., and F.R.T.; data curation, L.P.; writing—original draft preparation, L.P.; writing—review and editing, L.P. and R.D.V.; supervision, L.P. All authors have read and agreed to the published version of the manuscript.

**Funding:** This research received no external founds.

**Conflicts of Interest:** The authors declare no conflict of interest.

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
