# Peer review of "Optical Behavior and Surface Analysis of Dental Resin Matrix Ceramics Related to Thermocycling and Finishing"

_applsci, doi:10.3390/app12094346_

Round 1
Reviewer 1 Report
Dear Authors, congratulations for all your hard work and dedication. The article is well written and the results are meaningful for this topic. However, there are some issues which must be addressed:
- Line 33- please replace the word "unbelievable" with a more objective term
- Line 34- please revise the meaning of the phrase "offering practitioners the chance to achieve dental restorations"
- Line 52-53- please consider rephrasing "they lose their surface gloss"- "color stability in time more easy"- compared to what type of materials?
- Line 57- to improve the resistance to staining
- Line 59- a need for more studies
- Line 62- please consider replacing "Vickers evaluation indentation hardness" with "Vickers hardness test"
- Line 77- and thus preserve optical properties
- Line 101- please consider replacing "mimic" with "such as mimicking"
- Line 108- please consider replacing "Five resin-matrix CAD-CAM ceramic materials:" with " Five resin-matrix CAD-CAM ceramic materials were selected for the study:"
- Line 113- "The CAD/CAM blocks were sliced"- please define the slicing method and instruments used
- Line 116- please state why this procedure was performed and how did you standardize the polishing protocol? Also, the application of this Renfert polish all-in-one paste is not indicated
- Line 118- please state why the Resin Glaze from Shofu was chosen for this application? Why didn't you use the recommend glazing protocol indicated by each manufacturer?Also, please state why did you apply 2 coats of glaze?
- Line 193- "for" was written 2 times. Please delete one.
- Figure 1 and 2- can you increase the size of the images? they are difficult to read
- Line 281- after the comma, please consider adding "and it is known"
- Line 297- please consider replacing "gaze removing during thermocycling" with "glaze being removed during thermocycling".
Author Response
Thank You very much for your review!
Line 33- please replace the word "unbelievable" with a more objective term
Response:We agree to this point and the word was replaced with outstanding.
Line 34- please revise the meaning of the phrase "offering practitioners the chance to achieve dental restorations"
Response:We agree to this point and corrected.
Line 52-53- please consider rephrasing "they lose their surface gloss"- "color stability in time more easy"- compared to what type of materials?
Response: Well known resin composites wear out, they lose their surface gloss and color stability in time more easy compared to ceramic materials.
Line 57- to improve the resistance to staining
Response :We agree to this point in the review and corrected.
Line 59- a need for more studies
Response :We agree to this point in the review and corrected.
Line 62- please consider replacing "Vickers evaluation indentation hardness" with "Vickers hardness test"
Response :We agree to this point in the review and corrected.
Line 77- and thus preserve optical properties
Response :We agree to this point in the review and corrected.
Line 108- please consider replacing "Five resin-matrix CAD-CAM ceramic materials:" with " Five resin-matrix CAD-CAM ceramic materials were selected for the study:"
Response: We corrected accordingly.
Line 113- "The CAD/CAM blocks were sliced"- please define the slicing method and instruments used
Response: We agree to this point and we added more informations.
The CAD/CAM blocks were sliced in rectangular-shaped plates (n=16 per material using a machine (Orthoflex PI Dental, Budapest, Hungary)that provides millimeter accuracy.
Line 116- please state why this procedure was performed and how did you standardize the polishing protocol? Also, the application of this Renfert polish all-in-one paste is not indicated
Response:The protocol for polishing followed other articles in literature that included this type of materials. Renfert polish all in one was used because it has indication for the hybrid materials such as the ones included in the manuscript.
- Line 118- please state why the Resin Glaze from Shofu was chosen for this application? Why didn't you use the recommend glazing protocol indicated by each manufacturer?Also, please state why did you apply 2 coats of glaze?
Response:We choose Resin Glaze for Shofu for all the samples in order to provide standardisation and to follow the same protocol for all the materials in order to be able to compare them statistically.
- Line 193- "for" was written 2 times. Please delete one.
Response:It was corrected.
Figure 1 and 2- can you increase the size of the images? they are difficult to read
Response: We agree to this point and we changed the font size for the all the figures.
Line 281- after the comma, please consider adding "and it is known"
Response :We agree to this point and added ‚’’and’’.
Line 297- please consider replacing "gaze removing during thermocycling" with "glaze being removed during thermocycling".
Response:We agree to this point and replaced the expression.
Reviewer 2 Report
The authors present an in vitro study evaluating the optical behavior and surface characteristics of five dental resin ceramics before and after thermocycling. The samples were polished or glazed and the surface treatments were compared. Samples preparation is described, followed by the description of the analysis performed.
The English language correction needs correction throughout the manuscript. See, for instance, lines 11-12, 16, 55, 281.
Lines 18-19: you should refer to the statistical methods used and the p-value considered statistically significant and not “statistical analysis was performed”.
The introduction section would benefit from being shortened and rearranged. I suggest the authors' group the information related to materials changes in the oral cavity and after the explanation for such changes. Later, refer to the advantages of hybrid resins to overcome such disadvantages.
Lines 103-105: I suggest a more focused aim to be stated. For instance, to evaluate if the finishing procedure influences the surface characteristics of the materials. What is the null hypothesis tested?
Lines 114: how was the sample size determined?
Table 1: I suggest adding the shades used to the table.
Lines 129-133: I suggest (t1) be placed in line 132 after the cycle description, instead of line 130.
I suggest images be added to the manuscript to illustrate the performed analysis.
Section 2.5: I understand microhardness evaluation permanently deforms the samples, but why did you not evaluate a subgroup at t1, similar to the other evaluations performed?
Section 2.6: was the data normality/not normality evaluated before choosing the statistical test?
Figures 1 and 2, and tables 2,3 and 4: please add * or other symbols to indicate the statistically significant differences.
Why do authors think the correlation between TP and OP is weaker after thermocycling?
Lines 259-260: the authors state “glaze was removed by thermocycling”. How was this evaluated? Why is not described in the manuscript?
Lines 261-262: the authors state “glaze should be avoided”. However, in the introduction section (55-58) the advantages of glaze are presented. Do authors think this statement to avoid glazing is excessive?
Author Response
Thank You very much for the review!
The authors present an in vitro study evaluating the optical behavior and surface characteristics of five dental resin ceramics before and after thermocycling. The samples were polished or glazed and the surface treatments were compared. Samples preparation is described, followed by the description of the analysis performed.
The English language correction needs correction throughout the manuscript. See, for instance, lines 11-12, 16, 55, 281.
Response:We agree to this point and we corrected the English mistakes in the manuscript.
Lines 18-19: you should refer to the statistical methods used and the p-value considered statistically significant and not “statistical analysis was performed”.
Response:Statistical analyses were performed with IBM SPSS Statistics software at a significant value for p<0.05.
The introduction section would benefit from being shortened and rearranged. I suggest the authors' group the information related to materials changes in the oral cavity and after the explanation for such changes. Later, refer to the advantages of hybrid resins to overcome such disadvantages.
Response: We agree to this point and we arranged the introduction section.
Lines 103-105: I suggest a more focused aim to be stated. For instance, to evaluate if the finishing procedure influences the surface characteristics of the materials. What is the null hypothesis tested?
Response:We agree to this point and added the null hypothesis.
Lines 114: how was the sample size determined?
Response: The sample size was selected based on the confidence level and the margin of error. The sample size selection was also based on the Central Limit Theorem that justifies the use of a normal distribution if the sample size is large enough. It is said that a sample size equal to 30 or greater is accepted to conduct a study.
Table 1: I suggest adding the shades used to the table
Response:We agree to this point and we added the shades in the table.
Lines 129-133: I suggest (t1) be placed in line 132 after the cycle description, instead of line 130.
Response: We agree to this point and changed the term (t1).
I suggest images be added to the manuscript to illustrate the performed analysis.
Response: We agree to this point and we added another 3 images to ilustrate the performed analysis.
Section 2.5: I understand microhardness evaluation permanently deforms the samples, but why did you not evaluate a subgroup at t1, similar to the other evaluations performed?
Response: We agree to this point and we replaced the figure with the correct microhardness evaluation values.
Section 2.6: was the data normality/not normality evaluated before choosing the statistical test?
Response: The date was evaluated using the power test.
Figures 1 and 2, and tables 2,3 and 4: please add * or other symbols to indicate the statistically significant differences.
Response: We agree to this point and added informations in the tables.
Why do authors think the correlation between TP and OP is weaker after thermocycling?
Response: The correlation is weaker after themal aging because the optical properties are effected by ther thermal aging. The Pearson correlation was used.
Lines 259-260: the authors state “glaze was removed by thermocycling”. How was this evaluated? Why is not described in the manuscript?
Response: After thermal ging the glaze could be easily seen in the areas where it was removed.
Lines 261-262: the authors state “glaze should be avoided”. However, in the introduction section (55-58) the advantages of glaze are presented. Do authors think this statement to avoid glazing is excessive?
Response: We agree to this point and we considered the results of this study and after hydrothermal aging the glaze was partially removed and this is why we considered that for this type of materials the glaze should be avoided.

Reviewer 3 Report
The manuscript reports the effect of finishing and thermocycling on the optical properties of resin matrix ceramic materials. Considering the ever-increasing interest in the CAD/CAM technology, the results and conclusions obtained in this work may be beneficial for the dental industry. However, there are several problems in the manuscript. Therefore, the manuscript cannot be accepted in the current state, and is suggested for reevaluation after revision.
- The introduction (Section 1) contains a good summary of the recent works on the topic. However, there is only one sentence that describes their research (Line 103-105). The introduction should provide at least an overview of the paper’s structure, and preferably with a clear research questions/hypotheses.
- In Materials and Methods (Section 2), the authors did not explain the conditions of thermocycling such as the temperatures and the dwelling time.
- What is CR? Also, define Sp, Ep, Lp, Hp, Cp, Hg, Eg, Sg, Cg and Lg. Introduce every acronym before using it in the text.
- Tables 3, 4 and 5 have so many numbers but no visual illustration is provided. It is hardly understandable whether or not there is statistical significance. Add a couple of figures for a better readability.
- There are numerous English problems.
Author Response
Thank You very much for the review.
The manuscript reports the effect of finishing and thermocycling on the optical properties of resin matrix ceramic materials. Considering the ever-increasing interest in the CAD/CAM technology, the results and conclusions obtained in this work may be beneficial for the dental industry. However, there are several problems in the manuscript. Therefore, the manuscript cannot be accepted in the current state, and is suggested for reevaluation after revision.
The introduction (Section 1) contains a good summary of the recent works on the topic. However, there is only one sentence that describes their research (Line 103-105). The introduction should provide at least an overview of the paper’s structure, and preferably with a clear research questions/hypotheses.
Response:The study aimed to conduct a surface analysis of resin matrix ceramic materials related to finishing and thermocycling, and to assess their consequences on optical properties. The study included hybrid materials that were tested after hydrothermal aging for the changes in the optical properties such as translucency and opalescence, and as well for the surface and mechanical properties.
In Materials and Methods (Section 2), the authors did not explain the conditions of thermocycling such as the temperatures and the dwelling time.
Response:We agree to this point and other informations were added in text.
Each cycle lasted 80 s: 30 s was the dwelling time in the 5°C bath, 10 s needed for transfer to the other bath, 30 s was the time in the 55°C bath, 10 s to transfer of the samples back to the 5°C bath. Other 5000 cycles were followed using the same protocol (t2).
- What is CR? Also, define Sp, Ep, Lp, Hp, Cp, Hg, Eg, Sg, Cg and Lg. Introduce every acronym before using it in the text.
Response:We agree to this point and corrected in the manuscript.
Tables 3, 4 and 5 have so many numbers but no visual illustration is provided. It is hardly understandable whether or not there is statistical significance. Add a couple of figures for a better readability.
Response:We agree to this point and added other figures in the manuscript.
There are numerous English problems.
Response:We agree to this point and corrected.

Reviewer 4 Report
Proposed article deals with the optical and surface characteristics of dental resin matrix ceramics related to thermocycling and finishing. Introduction part should be expanded as it is not clear in what way the accelerated ageing of materials applied in dental prostheses are being carried.
In the materials and methods part separat paragraph should be aded regarding the detailed process of accelerated aging. In Table 1, the shade and translucency of the material should be added. Explain why you talk about 2 test, when the test 2 is the same as test 1 with increasing number of cycles. Why calculate mean value of roughness of 5 different directions.
What was measurement geometry for spectrophotometric measurements.
Define abbreviation CR before the equation (3).
In the section entitled Statistical analysis hydrothermal ageing was mentioned. I suggest using this term through the entire article instead of accelerated ageing.
In the Results section add the figure number in the text where you discuss about the results that are represented on the figure.
There are no result presented for colorimetric measurements. L*a*b* diagrams should be added.
Discussion should be expanded. Lines 300-305 should be moved to introduction.
Conclusion should be expanded. The novelty of the work and the impact for the future research should be added.
Author Response
Thank You very much for your review!
Proposed article deals with the optical and surface characteristics of dental resin matrix ceramics related to thermocycling and finishing. Introduction part should be expanded as it is not clear in what way the accelerated ageing of materials applied in dental prostheses are being carried.
In the materials and methods part separat paragraph should be aded regarding the detailed process of accelerated aging.
Response: We agree to this point and added another reference in the manuscript and another paragraph.
The hydrothermal aging protocol has indication for this type of dental materials. This method is used to simulate in vivo aging of the dental restorative materials by subjecting them to repeated cold and warm water baths[92]. These water baths simulate the changes that occur in the oral cavity of the patients.
In Table 1, the shade and translucency of the material should be added.
Response:We agree to this point and we added the shade and translucency in the manuscript.
Explain why you talk about 2 test, when the test 2 is the same as test 1 with increasing number of cycles. Why calculate mean value of roughness of 5 different directions.
Response: We calculated the roughness in 5 different directions to have a proper view of the surface differences that take place on the surface of the samples before and after themal aging.
What was measurement geometry for spectrophotometric measurements.
Response: We calculated the unitless parameters TP, OP and CR according to their formulas. The values for L*, a*, b* resulted after spectrofotometric analysis with Vita Easy Shade device.
Define abbreviation CR before the equation (3).
Response: We agree to this point and we added information in the manuscript.
In the section entitled Statistical analysis hydrothermal ageing was mentioned. I suggest using this term through the entire article instead of accelerated ageing.
Response:We agree to this point and added in the manuscript ‘’hydrothermal aging’’.
In the Results section add the figure number in the text where you discuss about the results that are represented on the figure.
Response:We agree to this point and added in text the figure number.
There are no result presented for colorimetric measurements. L*a*b* diagrams should be added.
Response: We agree to this point and added other 2 figures to ilustrate the tables. We calculated the parameters TP, OP and CR and used the . L*a*b* values in the formulas.
Discussion should be expanded. Lines 300-305 should be moved to introduction.
Response:We agree to this point and moved the paragraph in the introduction section.
Conclusion should be expanded. The novelty of the work and the impact for the future research should be added.
Response: We agree to this point and added another point in the conclusion section.
This research stands as a starting point for the future research because of the various hybrid materials included in this text and of the significant results. Taking everything into consideration, we can only encourage further investigations so that the results of different studies can be compared and analyzed properly.

Round 2
Reviewer 3 Report
Thanks for addressing all the points I have suggested in the revised manuscript.
Reviewer 4 Report
The manuscript was revised according to the suggestions.